# Effect of Electrolyte Concentration on the Electrochemical Performance of Spray Deposited LiFePO_4_

**DOI:** 10.3390/nano13121850

**Published:** 2023-06-13

**Authors:** Christina Floraki, Maria Androulidaki, Emmanuel Spanakis, Dimitra Vernardou

**Affiliations:** 1Department of Electrical and Computer Engineering, School of Engineering, Hellenic Mediterranean University, 71410 Heraklion, Greece; 2Institute of Electronic Structure and Laser (IESL), Foundation for Research and Technology Hellas (FORTH), 70013 Heraklion, Greece; pyrhnas@physics.uoc.gr; 3Department of Materials Science and Technology, University of Crete, 70013 Heraklion, Greece; spanakis@materials.uoc.gr; 4Institute of Emerging Technologies, Hellenic Mediterranean University Center, 71410 Heraklion, Greece

**Keywords:** spray deposition, LiFePO_4_, graphite, electrolyte concentration, electrochemical evaluation

## Abstract

LiFePO_4_ is a common electrode cathode material that still needs some improvements regarding its electronic conductivity and the synthesis process in order to be easily scalable. In this work, a simple, multiple-pass deposition technique was utilized in which the spray-gun was moved across the substrate creating a “wet film”, in which—after thermal annealing at very mild temperatures (i.e., 65 °C)—a LiFePO_4_ cathode was formed on graphite. The growth of the LiFePO_4_ layer was confirmed via X-ray diffraction, Raman spectroscopy and X-ray photoelectron spectroscopy. The layer was thick, consisting of agglomerated non-uniform flake-like particles with an average diameter of 1.5 to 3 μm. The cathode was tested in different LiOH concentrations of 0.5 M, 1 M, and 2 M, indicating an quasi-rectangular and nearly symmetric shape ascribed to non-faradaic charging processes, with the highest ion transfer for 2 M LiOH (i.e., 6.2 × 10^−9^ cm^2^/cm). Nevertheless, the 1 M aqueous LiOH electrolyte presented both satisfactory ion storage and stability. In particular, the diffusion coefficient was estimated to be 5.46 × 10^−9^ cm^2^/s, with 12 mAh/g and a 99% capacity retention rate after 100 cycles.

## 1. Introduction

Aqueous metal-ion (Li, Zn, Na, K, Mg, Ca, etc.) batteries and ammonium-ion batteries have exhibited extraordinary qualities for applications in energy storage, owing to their quality of security and the different electrode materials that can be utilized [1,2]. Lithium iron phosphate (LiFePO_4_) is an excellent cathode material for Li-ion batteries (LIBs) because it is extremely safe, thermally stable, and low cost [1,2]. Nevertheless, the electronic conductivity is poor, and the diffusion coefficient is slow, which limit the development for high power devices [3]. Hence, there is a lot of space for research to explore ways to improve the material’s performance. There is an increasing interest in the optimization of the synthesis route (doping modification, morphological regulation, nanosized particles) [4,5,6], the coating with electron-conducting layer and orientation control [7], and computational research on the understanding of the ionic dynamic properties of LiFePO_4_ [8]. However, there is not sufficient information on the investigation of LiFePO_4_ thin films as cathodes. Thin film manufacturing processes have significant advantages because the cost of scalable roll-to-roll processes is reduced. In addition, the thickness of the cathode can be altered to a value such that the deficient electronic conductivity does not affect the electrochemical performance of the electrode [9].

There are several reports on the growth of LiFePO_4_ film deposition, including pulsed laser deposition [10], radio frequency magnetron sputtering [11,12,13,14,15,16], sol–gel [17,18], and drop casting [19]. The majority of these methods require high processing costs and annealing temperatures (i.e., ≥500 °C). In this paper, a simple, multiple-pass deposition technique will be utilized in which the spray-gun will be moved across the substrate, creating a “wet film”, which—after thermal annealing at very mild temperatures (i.e., 65 °C)—will form a cathode material on the carbon component. The carbon component (1 × 1 cm^2^) is chosen as a substrate because it is expected to act as a support for uniformly anchoring the LiFePO_4_ film. In that perspective, graphite is a suitable material because of its mixed electron-ion conductivity and structural stability [20]. 

Most researchers have focused on the electrochemical performance of LiFePO_4_ in non-aqueous electrolytes, since they possess wide electrochemical stability windows that enable the working voltage ranges of the cathode and anode materials [21,22,23,24]. However, issues such as flammability, cost (i.e., LiPF_6_ in carbonate solvent [25]), toxicity, and safety are still unresolved. On the other hand, aqueous electrolytes have advantages including low cost, high safety, elevated conductivity, which reduces the equivalent series resistance and improves power distribution, and environmental friendliness [26,27]. Thus, the current investigation seeks to identify the appropriate electrolyte concentration of lithium hydroxide (LiOH) with affordable properties related with the cycling stability and the specific capacity. Our choice is based on the use of LiOH to both synthesize and electrochemically test LiFePO_4_ electrodes. 

Regarding the film manufacturing process, spray deposition will be our approach because it combines the advantages of low cost, low-complexity environments (i.e., in ambient air, low temperature processing, binder- and surfactant-free materials) and high throughput [28,29,30]. In particular, a water-based solution will be prepared via a simple method of mixing the main precursors. A spray technique will then be demonstrated to prepare the LiFePO_4_ coating on the graphite substrate as a thin film electrode at 65 °C. The best performing combination of electrode–electrolyte will be evaluated through the selective use of aqueous electrolyte at different concentrations (i.e., 0.5 M, 1 M, 2 M). 

## 2. Materials and Methods

### 2.1. Raw Materials

The materials used for the synthesis of LiFePO_4_ were lithium hydroxide (LiOH) powder (reagent grade, ≥98%), phosphoric acid (H_3_PO_4_, ≥97%), and iron(II) sulphate heptahydrate (FeSO_4_), all supplied by Sigma Aldrich (Merck SA 41-45 Kifisias av. (Building B) 15123 Marousi Athens, Greece). Regarding the electrolyte, LiOH was also utilized along with deionized water.

### 2.2. Spray Deposited LiFePO_4_

The solution was prepared by dissolving the correct amount of LiOH, iron sulphate (FeSO_4_), and phosphoric acid (H_3_PO_4_) in distilled water in a stoichiometric ratio of 1:1:1. Following this procedure, LiOH was initially mixed with FeSO_4_, with the final addition of H_3_PO_4_. The concentrations of Li^+^, Fe^2+^, and PO_4_^3+^ were all 0.01 M and the volume of the final solution reached 60 mL. Then, 10 mL of the prepared solution were placed in the spray gun for the LiFePO_4_ deposition on the graphite substrate (Figure 1). The graphite substrate, with a size of 1 × 1 cm^2^, was placed on a hot plate at 65 °C. The distance between the spray gun and the substrate was kept at 13 cm, moving the spray gun (to the right and left) across the substrate for the deposition to take place. For each spray, 10 s was allowed to elapse for the solution to dry, and the subsequent spray was continued. After the completion of the spraying process, the LiFePO_4_/graphite remained on the hotplate for 15 more minutes. 

### 2.3. Basic Characterization

X-ray diffraction (XRD) analysis was performed to study the structure of LiFePO_4_ using SmartLab^®^ SE (by Rigaku Europe SE- Hugenottenallee 167 Neu-Isenburg 63263, Germany) for processing parameters: power 46 kV, 50 mA, and speed time 8 min. Confocal micro-Raman measurements at room temperature were acquired with a system of Thermo Fisher Scientific model DXRxi. We used a 532 nm laser as the excitation beam with 20 mW power, exposure time 0.1 s, and objective lens long distance ×10. In addition, a monochromated Al-Kα source was utilized for X-ray photoelectron spectroscopy (XPS) measurements in FlexMod (SPECS- SPECS Surface Nano Analysis GmbH Voltastrasse 5, 13355 Berlin, Germany) with X-ray source XR-50 and 15 kV/200 W. Finally, field-emission scanning electron microscopy (FE-SEM) was used to analyze the morphology of the as-grown LiFePO_4_ in JSM-IT700HR InTouchScope™ Field Emission SEM (by Thermo Fisher Scientific- Neuhofstrasse 11, 4153 Reinach TechCenter, 4153 Basel, Switzerland) for processing parameters: 20 kV power, 10 μm width.

### 2.4. Electrochemical Evaluation of LiFePO_4_

For the electrochemical evaluation of LiFePO_4_ film cathodes, a three-electrode electrochemical cell was utilized [31,32]. The working electrode was the LiFePO_4_ film on graphite, the counter electrode was the graphite, and the reference electrode was Ag/AgCl. The measurements were performed in 0.5 M, 1 M, and 2 M aqueous solutions of LiOH with a scan rate of 50 mV/s and potential range of −0.7 V to +0.7 V. Measurements were also carried out at different scan rates of 10, 20, 30, 40, 50, and 100 mV/s. Finally, galvanostatic charge/discharge tests of LiFePO_4_ were performed at specific current 1.2 mA and ambient temperature (25 ± 1 °C). The electrochemical measurements took place in Autolab PGSTAT101 by Metrohm AG.

## 3. Results

### 3.1. Structure and Morphology Evaluation

Figure 2a presents the XRD pattern of spray deposited LiFePO_4_ film on graphite, revealing the simultaneous presence of LiFePO_4_ and substrate peaks. In particular, two low intensity peaks are indicated at 24.28° and 53.02° with Miller indices (011) and (222), respectively, for LiFePO_4_ [33], in contrast with the strong signals from graphite (i.e., at 26.6° and 54.5° corresponding to (002) and (004) Miller indices) [34]. This behavior is due to the background intensities caused by the scattering from the substrate. Figure 2b shows the Raman spectra of the graphite substrate and LiFePO_4_ film on graphite. The graphite spectrum is highly ordered, since it shows one in-plane vibration of the graphite lattice (G band) at 1575 cm^−1^ and a disorder band caused by the graphite edges at 1355 cm^−1^ [35]. Regarding the LiFePO_4_ [36,37], two peaks can be identified at 1005 cm^−1^ and 1092 cm^−1^, indicating the non-distorted PO_4_^3−^ tetrahedral in the pristine LiFePO_4_. The mode at 425 cm^−1^ is assigned to O-P-O bending internal to the PO_4_^3−^ anion. Finally, the mode at approximately 215 cm^−1^ is due to Fe-O vibrations.

Figure 3a shows the O 1s spectrum with two deconvoluted peaks at 531.5 eV and 534.1 eV, attributed to oxygen atoms of the PO_4_^3−^ phosphate groups in LiFePO_4_ [38], and oxygenated species adsorbed on the electrode surface deriving possibly from electrolyte degradation [39], respectively. In P 2p spectrum (Figure 3b), two peaks are observed at 132.3 eV and 133.4 eV, which are fitted to 2p_3/2_ and 2p_1/2_ components, respectively, due to PO_4_^3−^ phosphate group indicating that this is the only phosphorus environment [40,41]. Li 1s spectrum (Figure 3c) shows one deconvoluted peak at 54.4 eV assigned to LiFePO_4_ [42]. Finally, the Fe 2p spectrum (Figure 3d) shows two peaks at 710.5 eV and 722.4 eV, which correspond to 2p_3/2_ and 2p_1/2_ for Fe^3+^ [43]. There is one additional peak at 714.7 eV, which is characteristic of Fe^2+^ with a weaker signal than Fe^3+^ [42,44]. The presence of this impurity may be due to the air exposure of the LiFePO_4_ material. All the above analysis, including XRD and Raman spectroscopy, confirm the presence of LiFePO_4_.

The surface morphology of the LiFePO_4_ film grown on graphite is presented in Figure 3e. Figure 3e indicates a thick LiFePO_4_ film consisting of FePO_4_ flake-like particles with large size distribution (200 nm to 1 μm) observed [44]. A similar microstructure was also indicated for LiFePO_4_ prepared by a high-energy balling system [45] and chemical fabrication [46]. This type of morphology is expected to positively affect the electrochemical performance of the material under investigation because of the high contact area between the electrolyte and the cathode favoring the Li^+^ diffusion. 

### 3.2. Electrochemical Analysis

In order to find the appropriate electrolyte concentration, the cyclic voltammetry (CV) curves were evaluated in 0.5 M, 1 M, and 2 M LiOH electrolytes (Figure 4a–c). The curves exhibit an almost rectangular shape, with two inconspicuous peaks in the redox processes (i.e., at approximately −0.5 V (cathode) and +0.3 V (anode)) indicating a non-faradaic charging process [47]. This process is based on the formation of a double layer at the electrode–electrolyte interface during the adsorption of Li^+^ on LiFePO_4_ film surface, as proposed in Equation (1) [47,48,49].
LiFePO_4_ + Li^+^ + e^−^ <-> (LiFePO_4_-Li^+^)_surface_(1)

In that case, the charge is mainly stored in the electrolyte and the electrolyte concentration is therefore expected to affect the cathode’s performance [50]. The effect of LiOH concentration was studied, keeping the scan rate (i.e., 50 mV/s) and the potential window (i.e., −0.7 V to +0.7 V) constant for different scan numbers. In aqueous electrolytes with high salt concentration (2 M), the ion transfer is larger, resulting in higher specific current as confirmed in Figure 4c. The ionic conductivity of LiOH for different concentrations is illustrated in Figure 3f, indicating that the enhanced electrochemical performance in 2 M LiOH is attributed to the high conductivity [51]. However, the stability of the cathode in highly concentrated LiOH electrolyte is poor after 100 scans, as one can observe from Figure 4c, which is also confirmed from the peeling of the sample in the electrochemical cell. In order to substantiate this performance, the specific capacity for each scan number was calculated from the cyclic voltammograms using Equation (2) [52]
C = ∫Idv/(2 × 3.6 × m × v)(2)
where ∫Idv is the area of the CV curve, m is the mass of the active material in g, and v is the scan rate in V/s. In addition, the percentage change of specific capacity was estimated from Equation (3) where final is the specific capacity at 100 scans and initial is the respective value for the first scan.
(3)$$\%\mathrm{change}=\frac{\mathrm{final}-\mathrm{initial}}{\mathrm{initial}}\times 100\%$$

The % change was found to be 16% (0.5 M LiOH), 11% (1 M LiOH), and 45% (2 M LiOH), verifying the enhanced stability of LiFePO_4_ film tested in 1 M LiOH aqueous electrolyte. The ion mobility reduction with time under strong alkaline conditions is not in agreement with the results reported by Luo et al., who suggest that LiFePO_4_ can be used over a range from 7 to 14 in aqueous solutions [53]. In that perspective, we could consider for future work the pH adjustment, the elimination of O_2_ (placing the electrochemical cell in a glove box), and the coating (e.g., TiO_2_) on the top of LiFePO_4_ as a protective layer.

The performance of the LiFePO_4_ film was also studied for scan rates of 10, 20, 30, 40, 50, and 100 mV/s in the different LiOH electrolyte concentrations (Figure 4d–f). All curves indicate the almost rectangular shape. In all cases, the specific current increases with the scan rate and the shape of the curves remains unchanged, demonstrating an excellent behavior for the LiFePO_4_ film electrode. Figure 4g–i present the variation of specific capacity as estimated from Equation (2), with scan rate for the different electrolyte concentration investigated showing a decreasing trend for higher scan rates. This is due to the fact that the fast scan rates do not give sufficient time to the ions to intercalate into the LiFePO_4_ film, resulting in lower specific capacities [54,55,56]. 

Based on Randles–Sevcik Equations (4) and (5) [55]
(4)$$I_{p}=D^{1/2}2.72\times 10^{5}n^{3/2}AC\nu^{1/2}$$
(5)$$D^{1/2}=\frac{a}{2.72\times 10^{5}n^{3/2}AC}$$
where *I*_p_ is the peak current in *A*, *n* is the number of electrons involved in the process, *A* is the area of the cathode in cm^2^, *D* is the diffusion coefficient in cm^2^/s, *c* is the concentration in mol cm^−3^, *v* is the scan rate in V s^−1^, and *a* is the slope as obtained in Figure 5 (left). To estimate the diffusion coefficient of the sample, the plot of the peak current as a function with the square root of the scan rate (υ^1/2^) (Figure 5 (right)) is initially obtained for the determination of the slope in each LiOH concentration. Following this procedure, the values are substituted on Equation (5) for the calculation of the diffusion coefficient. The highest value was 6.2 × 10^−9^ cm^2^/s for the 2 M LiOH, which can be attributed to the highest ionic conductivity facilitating electron transfer within the cathode material and contributing to its overall enhanced performance [57,58]. Based on Table 1, one can also realize that it is one of the highest values reported in the literature, possibly due to the appropriate combination of electrode–electrolyte characteristics. Since LiOH is an electrolyte that has not been studied extensively for LiFePO_4_, it is worth investigating it further through the careful addition of other salts, such as Li_2_SO_4_ [59], acting as conductive additives to optimize its conductivity.

The power law model can be used to estimate the charge storage mechanism. The peak current and the scan rate follow the power law as shown in Equations (6) and (7) [56]
i = av^b^(6)
log(i) = blog(v) + log(a)(7)

Figure 5 (right) presents the variation of log (peak current) as a function with log(scan rate) and the fitted lines of LiFePO_4_ studied in 1 M and 2 M LiOH aqueous electrolyte. The data obtained from the 0.5 M electrolyte were not further studied due to the general low performance. The slope of the fitted line is the b-value. If it is equal to 0.5, the process is diffusion-controlled, while for the case of 1.0, the surface-induced capacitive process is valid [60,61,62]. In this work, the b-value was estimated to be 0.67 for both cases, which is very close to 1.0, exhibiting a domination of non-faradaic process (i.e., Li^+^ adsorption on the surface of LiFePO_4_ film).

Figure 6 presents the galvanostatic charge/discharge tests in the absolute potential range between −0.7 V to +0.7 V (vs. Ag/AgCl) using graphite as a counter electrode in 0.5 M, 1 M, and 2 M LiOH aqueous electrolytes. The highest specific capacity of the cathode studied in 2 M LiOH is expected, as discussed above, due to the higher conductivity of the electrolyte. One can observe plateaus during the discharging process, which may be due to the electrochemical properties of the cathode rising between a combination of Li^+^ intercalation/deintercalation into the LiFePO_4_ and the adsorption on the cathode surface as supported from power law. The cathode evaluated in 1 M LiOH aqueous electrolyte presented a specific capacity of 12 mAh/g with a capacity retention rate of 99% after 100 cycles, as estimated from the difference between the specific capacity at 100 scans and the first scan divided by the specific capacity at the first scan. The curves after 100 cycles are not included since they coincide with those of the first scan.

Additionally, in Figure 6, a FE-SEM of spray deposited LiFePO_4_ film on graphite utilizing 10 mL spraying solution after cycling is shown. It presents the flake-like behavior, along with particle agglomerations and some cracks [63,64]. This is probably due to the volume changes taking place during the cycling process. 

Regarding the Raman spectra of LiFePO_4_ after cycling, there are FePO_4_ Raman modes, which are similar to LiFePO_4_. However, delithiation of LiFePO_4_ or lithiation of FePO_4_ leads to changes in both peaks’ amplitude and position [34]. Specifically, the G, D bands peaks are in lower wavenumbers (i.e., 1346 cm^−1^, 1588 cm^−1^) as those presented in Figure 2. There are three peaks at 946 cm^−1^, 1025 cm^−1^, 1061 cm^−1^ corresponding to asymmetric stretching of PO_4_^3−^, which appears due to the formation of FePO_4_ after the delithiation process. The lower peaks at 164 cm^−1^ and 387 cm^−1^ indicate the Fe-O and O-P-O bonds, as before cycling. From the above results it is confirmed that lithium ions insert in LiFePO_4_ during the lithiation process and that FePO_4_ is the second phase that is present on the delithiation from LiFePO_4_. The extraction of lithium ion from LiFePO_4_ to charge the cathode is presented as reaction (8), and the insertion of lithium into FePO_4_ on discharge as reaction (9) [65,66].
LiFePO_4_ + → xFePO_4_ + (1 − x)LiFePO_4_ + xLi^+^ + xe^−^(8)
FePO_4_ + xLi^+^ + xe^−^ → xLiFePO_4_ + (1 − x)FePO_4_(9)

The specific capacity of the cathode reported in this work is higher than the solid-state reaction process [54] and hydrothermal growth [60], while it is lower (one order of magnitude) than the direct recovery of scrapped LiFePO_4_ [4], the commercial powder [48], the mechanochemical activation of LiFePO_4_ [59], along with sol–gel [63] and spray-drying of LiFePO_4_/C [62] (Table 2). Overall, the growth methods utilized for the deposition of LiFePO_4_ are not practically feasible on a large scale for commercial applications. In that perspective, the combination of liquid-based chemistry with spray-coating can result in high quality films at a low temperature of approximately 65 °C, as indicated from the cathode characterization. In particular, it combines the advantages of low cost and low-complexity environments (i.e., in ambient air, low temperature processing, binder- and surfactant-free materials). Nevertheless, there is space for future work, including the involvement of a conductive material as a suspension in the spraying solution of LiFePO_4_ to enhance the cathode’s conductivity and as a consequence the overall electrochemical performance. 

Last but not least, restrictions for scaling-up results can be overcome through computational fluid dynamics (CFD) studies of the spray-gun process as a prospective work. Theoretical predictions of the lab-scale experimental process will be directly compared with experimental measurements to validate the developed computational model. Upon its validation, the model will be applicable for experimental set-ups and conditions corresponding to the scaled-up process.

## 4. Conclusions

A simple, multiple-pass deposition technique was utilized after thermal annealing at very mild temperatures (i.e., 65 °C) for the growth of a LiFePO_4_ layer on graphite as a cathode. The growth of the LiFePO_4_ layer was successfully confirmed via XRD, Raman spectroscopy, and XPS. When the cathode was tested in different LiOH concentrations, the highest electrolyte concentration resulted in an enhanced electrochemical performance due to its high conductivity, with, however, poor stability strengthening the importance of 1 M LiOH. The behavior of the LiFePO_4_ film was evaluated for different scan rates ranging from 10 mV/s to 100 mV/s, showing an excellent performance of the cathode with an almost rectangular shape of the CV curves and an increasing specific current with the scan rate. The specific capacity decreased with increasing scan rate, demonstrating that fast scan rates do not give sufficient time for the ions to intercalate into the LiFePO_4_ film, resulting in lower specific capacities. Overall, the cathode electrode studied in an aqueous solution of 1 M LiOH showed a specific capacity of 12 mAh/g with a capacity retention rate of 99% after 100 cycles and a diffusion coefficient of 5.46 × 10^−9^ cm^2^/s. This work gives a good basis and promising results for the future, focusing on the increase in the specific capacity of the cathode through pH adjustment (i.e., electrolyte solution), coating of a protective layer on the top of LiFePO_4_, and a controlled environment for the electrochemical evaluation to avoid the changes that may occur to the electrolytes, such as the possible conversion of LiOH to Li_2_CO_3_. From that perspective, further cycles need to be carried out along with structural/morphological analysis to understand the Li^+^ intercalation mechanisms.

## Figures and Tables

**Figure 1 nanomaterials-13-01850-f001:**
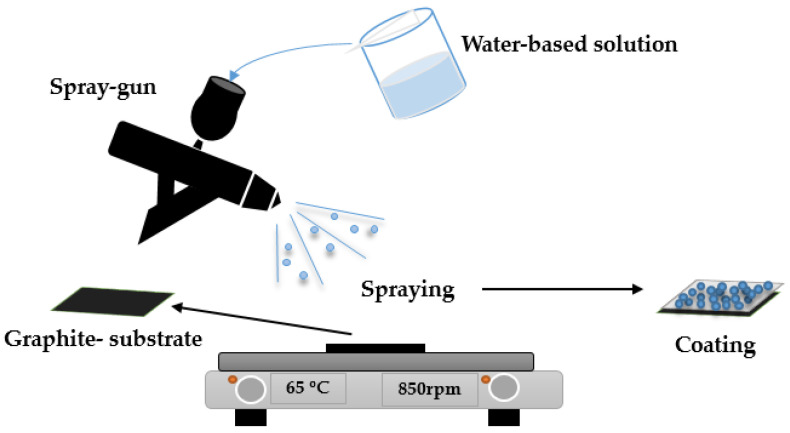
Schematic presentation of the spray deposition process.

**Figure 2 nanomaterials-13-01850-f002:**
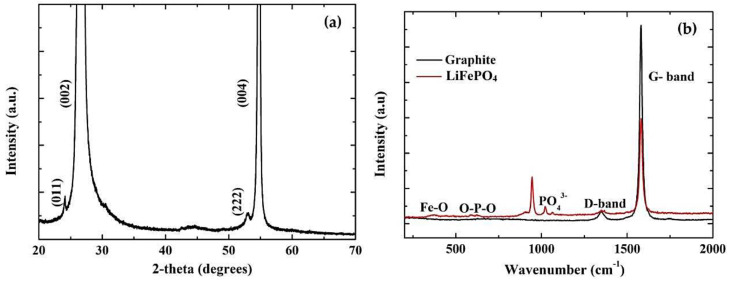
XRD of as-deposited LiFePO_4_ film on graphite (**a**) and Raman spectra of LiFePO_4_ film on graphite and graphite substrate (**b**).

**Figure 3 nanomaterials-13-01850-f003:**
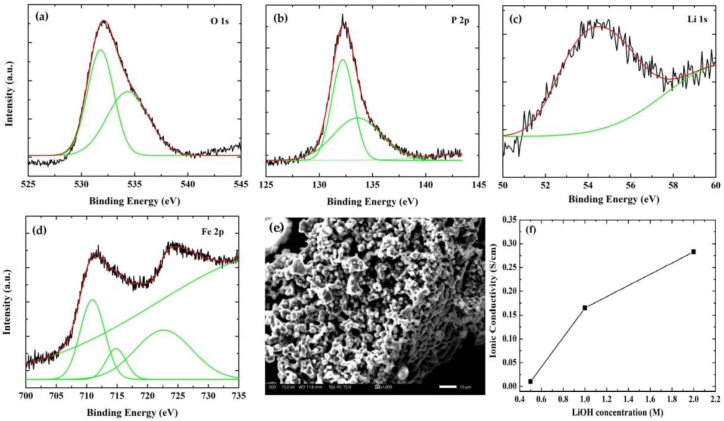
XPS spectra of O 1s (**a**), P 2p (**b**), Li 1s (**c**), Fe 2p (**d**) and FE-SEM of spray deposited LiFePO_4_ film on graphite utilizing 10 mL spraying solution (the bar is equal to 10 μm) (**e**), Ionic conductivity of LiOH different concentrations (**f**).

**Figure 4 nanomaterials-13-01850-f004:**
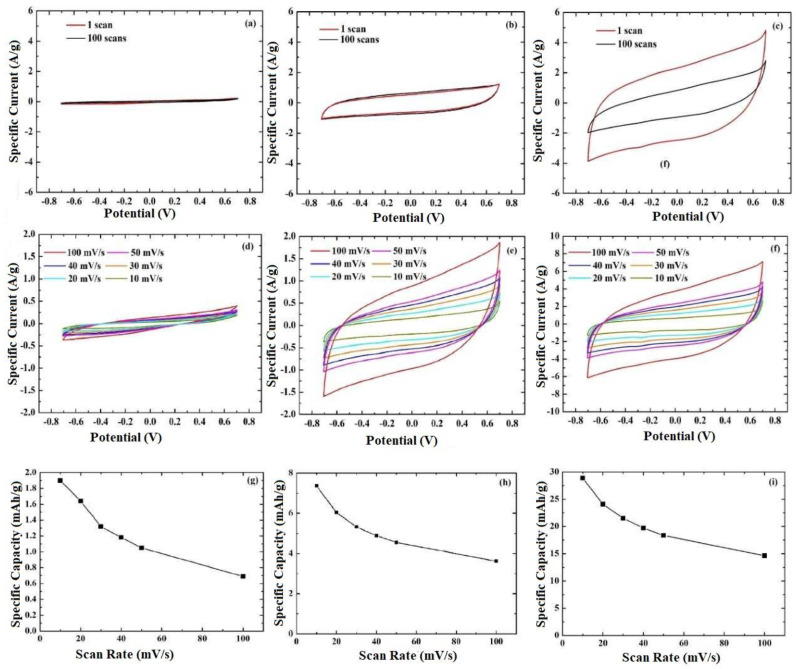
Cyclic voltammograms of the spray deposited LiFePO_4_ on graphite utilizing 10 mL spraying solution for scan rate 50 mV/s in 0.5 M (**a**), 1 M (**b**) and 2 M (**c**) LiOH aqueous electrolyte. Cyclic voltammograms of the same sample for different scan rates 10 mV/s, 20 mV/s, 30 mV/s, 40 mV/s, 50 mV/s and 100 mV/s varying the electrolyte concentration (**d**–**f**). Variation of specific capacity with the scan rate for each electrolyte concentration studied (**g**–**i**).

**Figure 5 nanomaterials-13-01850-f005:**
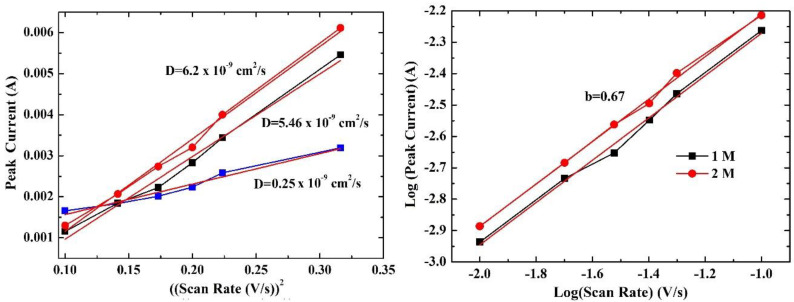
Variation of peak current with square root of scan rate for the LiFePO_4_ cathode studied in 0.5 M (blue line), 1 M (black line), and 2 M (red line) LiOH aqueous electrolyte for the estimation of diffusion coefficient (**left**). Variation of log(peak current) with log(scan rate) for the calculation of b-value for the cathode studied in 1 M and 2 M LiOH aqueous electrolyte (**right**).

**Figure 6 nanomaterials-13-01850-f006:**
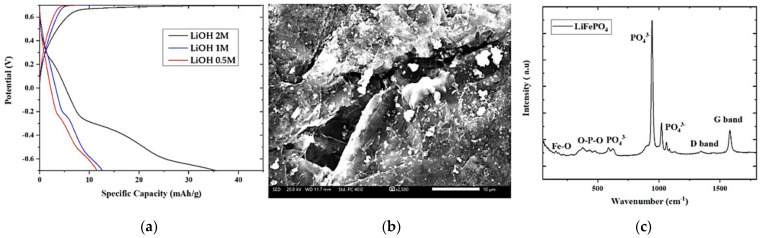
Charge/discharge curves of LiFePO_4_ in 0.5 M, 1 M and 2 M LiOH aqueous electrolytes for a potential range of −0.7 V to +0.7 V at 1.3 A/g. (**a**) FE-SEM of spray deposited LiFePO_4_ film on graphite utilizing 10 mL spraying solution after CV (**b**) and Raman spectra of LiFePO_4_ film on graphite and graphite substrate after CV measurements (**c**).

**Table 1 nanomaterials-13-01850-t001:** Diffusion coefficient values of LiFePO_4_ reported in the literature and this work as studied in aqueous electrolytes.

Cathode	Diffusion Coefficient (cm^2^/s)	Aqueous Electrolytes
LiFePO_4_/C (gel-combustion synthesis) [59]	0.8 × 10^−14^	Saturated LiNO_3_
LiFePO_4_ (commercial powder) [60]	2.020 × 10^−9^	Saturated LiNO_3_
LiFePO_4_ (in situ synthesis technique) [61]	1.5 × 10^−11^	1 M Li_2_SO_4_
LiFePO_4_/C (spraying drying process) [62]	1.22 × 10^−14^	0.5 M Li_2_SO_4_
LiFePO_4_ on Graphite	0.25 × 10^−9^	0.5 M LiOH This work
LiFePO_4_ on Graphite	5.46 × 10^−9^	1 M LiOH This work
LiFePO_4_ on Graphite	6.20 × 10^−9^	2 M LiOH This work

**Table 2 nanomaterials-13-01850-t002:** Specific capacity values of LiFePO_4_ cathode materials in aqueous electrolytes.

Cathode	Specific Capacity (mAh/g)	Aqueous Electrolytes
LiFePO_4_ (solid-state reaction process) [54]	13.3	1 M Li_2_SO_4_
LiFePO_4_ (hydrothermal growth) [60]	2.8 1.75	1 M KOH 2 M NaOH
LiFePO_4_ (direct recovery of scrapped LiFePO_4_) [4]	134	1 M Li_2_SO_4_
LiFePO_4_ (commercial powder) [48]	110	2 M Li_2_SO_4_
LiFePO_4_ (mechanochemical activation) [59]	130	0.5 M Li_2_SO_4_
LiFePO_4_/C (sol–gel) [63]	163.5	1 M Li_2_SO_4_
LiFePO_4_/C (spray-drying) [62]	140	0.5 M Li_2_SO_4_
LiFePO_4_ (spraying deposition)	36	2 M LiOH This work
LiFePO_4_ (solution method) [64]	65	1 M LiOH

## Data Availability

Not applicable.

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
