# Peer review of "Effect of Electrolyte Concentration on the Electrochemical Performance of Spray Deposited LiFePO4"

_nanomaterials, 2023, doi:10.3390/nano13121850_

Round 1

Reviewer 1 Report

The article entitled “Effect of Electrolyte Concentration on the Electrochemical Performance of Spray Deposited LiFePO4” submitted to Nanomaterials/MDPI has been reviewed. Energy storage technology is one of the most critical technologies for the development of new energy electric vehicles and smart grids. Among the energy storage technologies, lithium-ion technology coupled with abundantly available Fe-based cathode is an attractive energy storage device exhibiting excellent efficiency of charge-discharge processes along with high energy density. The electrolyte, a key component for the successful operation of energy materials, is greatly affected by its concentration. The electrolyte will affect the transport properties of Li+ ions as well as the solid interface film on the electrodes, and the concentration plays an important role in determining cell performance. The manuscript is fine with good electrochemical characterization. The work in its present form is publishable but needs some revisions (certain items must be clarified) before rendering a final decision.  The spray pyrolysis and deposition technique with thin films could be new to the readers.

The following points need to be considered.

·         In the abstract, line 17 – please mention the different LiOH concentrations in numbers like 0.5M, 1M, 2M, etc.

·         Abstract, line 18, is a quasi-rectangular and nearly symmetrical shape.

·         Line 20, please quantify the ion storage and capability.

·         How was the diffusion coefficient estimated through any formula?

·         Introduction, lines 37 – 38 a bit unclear. Please re-write.

·         What is the size of the carbon component substrate?

·         Lines 54 – 58 aqueous electrolytes have been introduced without any references. This is very weird.  Please include and discuss.

·         In the last paragraph of the introduction, please stress the novelty of the submitted work, like the effect of LIOH concentrations, using graphite substrate, thin film electrode, etc. Currently, it is weak.

·         Does the method explained in Section 2.1 scalable for practical purposes?

·         Why the XRD peaks are overshooting the plot?

·         The SEM morphology is not showing any particle size, or grains, why?

·         Is the current paper and LIFEPO4 reported for battery or super caps?

·         Over a period of time, LIOH will be converted to Li2CO3 by taking the atmospheric CO2. How the authors mitigated this?

·         Line 238; 242; Were there any cycling measurements shown?

·         In Table 5, the following key papers on LFP can be included such as doi.org/10.1016/j.ceramint.2022.03.043; and doi.org/10.1016/j.electacta.2010.09.011.

·         Figure 6b, are the potential values absolute (or) relative to any standard electrode?

·         What is the optimized electrolyte concentration concluded from this study?

Some minor edits will do.

Reviewer 2 Report

In this paper, LiOH was used as electrolyte to form LiFePO4 cathode on graphite to explore the influence of electrolyte concentration on electrochemical performance, and the work done by other researchers was summarized. In conclusion, this paper has important reference value for the researchers of electrolytes. Therefore, I suggest that this manuscript be published in nanomaterials, but the author needs to make some modifications before acceptance. The detailed comments are as follows:

1. Under the current of 1.3A/g, it is necessary to carry out a long cycle test on the battery to reflect its cycle stability, add charge and discharge curves with different cycles (such as the 20th, 50th and 100th cycles), and provide SEM before and after the cycle for comparison.

2. The purity of chemicals and reagents should be added to the Materials and Methods section.

3. The model and manufacturer of the characterization equipment should be added to the test section.

4. References should be carefully checked, such as references 40, 60, etc. The following papers may be helpful. Rare Met. 2021,40(12):3477―3484 (https://doi.org/10.1007/s12598-021-01783-4);Separation and Purification Technology 282 (2022) 120065; Int J Energy Res. 2021,1–12.

Reviewer 3 Report

In this manuscript, authors studied the effects of electrolyte concentration on the electrochemical performance of spray deposited LiFePO4. In general, it is an interesting work and the manuscript is well organized. However, there are still some issues to be addressed. A moderate revision is suggested before its acceptance.

1.     One or two sentences are required at the beginning of abstract to present the background or aim of this work.

2.     More introduction of the different energy storage sources should be provided with some more recent supporting articles, such as aqueous ammonium-ion batteries (Chemical Engineering Journal, 2023, 458, 141381); Li-ion battery (New Journal of Chemistry 45, 19446-19455, 2021); Zn-air battery (Molecules 28 (5), 2147, 2023); etc.

3.     More experimental details are suggested to be added into the figure 1.

4.     One more subsection on the raw materials should be added.

5.     The scale bar should be rebuilt to have a better readability.

6.     In Figure 4, all sub images should be added with the title for x and y axis. In addition, the images should be modified to have a better readability, especially the small texts.

7.     More background on the structure, properties and applications on electrolytes should be provided with supporting articles: Molecules 28 (5), 2042, 2023; Composites Communications 19 (239), 239-245, 2020; etc.

8.     The text style in table 1 should be modified with the same as main texts.

9.     There are too many too old references, which is better to be deleted or replaced with recent articles to show the novelty of this work.

10.  There are still some typos and grammar issues in the manuscript. Authors should carefully recheck the whole manuscript.

Round 2

Reviewer 1 Report

This reviewer went through the revised manuscript and the responses made by the authors. It has been fairly addressed, however, some minor issues must be clarified before going for publication. I have listed below:

1. In the abstract line 23, what is "99% change of the specific capacity"? It is a bit unclear. 

2. In the introduction, line 60, aqueous solutions are also very safe to operate.

3. In Results and Discussion, line 260 "with a 99% change after 100 cycles"  is unable to get the intended meaning.

4. A few typos must be carefully checked in the list of references.

Some edits are required; the intended meaning is unclear.

Author Response

Response 1: 

In line 255, we rewrote the sentence to make it clear.

''The cathode evaluated in 1 M LiOH aqueous electrolyte presented a specific capacity of 12 mAh/g with a capacity retention rate of 99 %  after 100 cycles as estimated from the difference between the specific capacity at 100 scans and the first scan divided by the specific capacity at the first scan. The curves after 100 cycles are not included since they coincide with those of the first scan.''

Response 2

We added the safety operation. Please see the manuscript

Response 3

It means capacity retention rate of 99% after 100 cycles. Changed.

Response 4

A few typos and the format of the references rechecked and changed.

Reviewer 3 Report

Authors have addressed most of the issues well except one:

Authors should carefully recheck the information of  references to make sure fulll and right information is provided, such as Ref. 1 should be "Wen, X., Luo, J., Xiang, K., Zhou, W., Zhang, C., Chen, H. (2023). High-performance monoclinic WO3 nanospheres with the novel NH4+ diffusion behaviors for aqueous ammonium-ion batteries. Chemical Engineering Journal, 458, 141381". Please also recheck others.

Author Response

Response :

Thank you, we rechecked the format and the information of the references carefully.